# Why Motor Competence Matters: Fundamental Movement Skills and Their Role in Promoting Physical Activity and Health in Czech Children Aged 9–10 Years

**DOI:** 10.3390/jfmk10030258

**Published:** 2025-07-08

**Authors:** Jakub Kokstejn, Miroslav Grobar, Jindrich Vampola, Martin Musalek

**Affiliations:** 1Department of Sport Games, Faculty of Physical Education and Sport, Charles University, Jose Martiho 269/31, 16252 Prague, Czech Republic; miroslav.grobar711@student.cuni.cz (M.G.); jindrich.vampola554@student.cuni.cz (J.V.); 2Department of Social Science Foundation in Kinanthropology, Faculty of Physical Education and Sport, Charles University, Jose Martiho 269/31, 16252 Prague, Czech Republic; martin.musalek@ftvs.cuni.cz

**Keywords:** motor skills, health-related variables, accelerometry, children, healthy status

## Abstract

**Background:** Motor competence (MC) plays a critical role in shaping children’s physical activity, health-related fitness, and weight status. However, few studies have explored these interrelationships during middle childhood. **Objectives:** The current study aimed to examine the relationships between MC and physical activity (PA), health-related fitness (HRF), and weight status (WS) in children during middle childhood (9–10 years). Simultaneously, the study aimed to find possible differences in PA, HRF, and WS between children with different levels of MC. **Methods:** In this cross-sectional study, the TGMD-2 (MC), UNIFITTEST 6–60 (HRF), ActiGraph GT3X (PA), and anthropometry (WS) were administered to 86 children (59 boys, age range of 9–10 years and mean age of 10.1 ± 0.6 years). **Results:** A significant strong association was observed between MC and HRF (*p* < 0.01, Cramer‘s V = 0.42). Significant moderate associations were observed between MC and WS (*p* < 0.05, Cramer’s V = 0.34) and between MC and PA (*p* < 0.05, Cramer‘s V = 0.25). Children with high MC proficiency demonstrated significantly higher levels of HRF (*p* < 0.01; large ES) and PA (*p* < 0.05; moderate ES) and a healthier WS (*p* < 0.01; moderate ES) than those with low MC proficiency. Similarly, children with moderate MC proficiency outperformed children with low MC in terms of HRF and WS. **Conclusions:** The strong associations between MC and key health indicators such as PA, HRF, and WS suggest that movement proficiency in middle childhood should be viewed as a cornerstone of health promotion. Although no causal claims can be made, the results emphasize the practical importance of integrating targeted motor skill development—especially FMSs—into school-based physical education and extracurricular programs. Supporting MC at this age may be a decisive step toward fostering long-term engagement in physical activity and improving children’s overall health trajectories.

## 1. Introduction

The situation surrounding children’s health is becoming critical worldwide, based on current research findings on physical activity (PA) and related health determinants. These studies confirm a dramatic decline in PA and health-related fitness (HRF) and, conversely, an increase in sedentary activities, a high incidence of childhood obesity, and several other modern lifestyle-related diseases [1,2,3]. Similarly, data from the most recent national survey in the Czech Republic, “Active Healthy Kids—Czech Republic 2022”, concluded that 42% of children are not sufficiently active compared with European standards for PA [4]. Furthermore, Czech children’s aerobic fitness levels ranked below the international average, falling at the 42nd percentile [4]. As a result of all these negative findings, considerable effort has recently been focused on identifying key determinants that significantly impact increasing PA, considered an essential health promotion and disease prevention strategy [5].

Motor competence (MC), primarily reflected through fundamental movement skills (FMSs), is widely recognized as a key determinant of physical activity (PA), which in turn can positively influence other health-related variables in children—such as health-related fitness (HRF) and weight status (WS)—especially during early-to-middle childhood [6,7,8,9,10]. These studies collectively highlight how MC supports engagement in PA. For instance, Stodden et al. [10] proposed a developmental model in which motor skill competency fosters PA and HRF, while Utesch et al. [9] demonstrated age-related strengthening of these relationships. Holfelder and Schott [8] further identified robust cross-sectional links between MC and structured PA, reinforcing the role of motor proficiency as a gateway to broader health benefits. MC in the form of FMSs serves as a comprehensive global term encompassing purposeful, large-scale movements involving major muscle groups or the entire body, as observed in physical activities like running, jumping, and maintaining balance [11]. During childhood, children begin acquiring FMSs, which serve as the cornerstone for mastering more advanced and complex movement skills in the future. FMSs are typically divided into locomotor skills (e.g., running, galloping, and hopping), manipulative or object-control skills (e.g., throwing, catching, kicking, and rolling), and balance or stability skills (e.g., body rolling, one-footed balance, stretching, and twisting) [12]. An adequate level of proficiency in FMSs should be obtained by age seven as children enter organized sports with specialized movement skills [12]. Learning through deliberate practice—including quality instruction, practice opportunities, and feedback—is a prerequisite for mastering skills at the required age [8,12]. However, current research regarding children’s motor competence has long reported low levels of FMSs among children [13,14,15]. Based on a comprehensive review by Bolger et al. [13], children, particularly in early middle childhood (6–8) and middle childhood (9–10), have demonstrated below-average FMS levels worldwide, while during early childhood, children (3–5 years) have reported average FMS levels. According to Bolger et al. [13], this negative trend concerning increasing age could have been the result of a limited quality of instruction and feedback or insufficient opportunities for FMS practice to improve their FMS levels. Another possible explanation is the transition from kindergarten to the first grade of primary school, where children become less physically active because of the more sedentary character of the teaching style in primary school [16].

The important role of MC in multiple health-related aspects of child development and sport participation has been extensively explained through a progression of conceptual models of and metaphors about motor development [7,10,12,17,18,19]. Early frameworks such as Seefeldt’s concept of a “proficiency barrier” [19] suggested that children with low MC are less likely to engage in PA and may struggle to develop more complex skills over time. Similarly, Clark and Metcalfe [18] introduced the metaphor of a “mountain of motor development”, illustrating the layered and lifelong progression of motor skill acquisition, with early MC forming the foundational base. Building upon these ideas, Stodden et al. [10] proposed a pivotal model that highlighted dynamic and reciprocal relationships between MC, PA, HRF, and WS. In this framework, MC positively influences PA, which subsequently enhances HRF and contributes to a healthier WS, while the WS can in turn affect each of these components. These interactions are theorized to strengthen with age, especially during middle-to-late childhood (9–12 years), as children’s increasing motor proficiency enables them access to more varied and complex physical activities [2]. More recent models have evolved beyond the purely physical dimension of motor development. Gallahue et al. [12] introduced a developmental taxonomy emphasizing the importance of mastering motor skills during early childhood as a foundation for lifelong movement competence. Hulteen et al. [17] extended the traditional FMS framework by proposing the broader concept of “foundational movement skills”, which includes essential movement capacities such as swimming, cycling, squatting, and lunging, skills which are important for physical literacy but often omitted from standard FMS assessments. Cairney et al. [7] further expanded this perspective by embedding MC within a holistic model of physical literacy that incorporates not only physical but also cognitive, emotional, and social dimensions. Their model underscores the importance of psychosocial factors such as motivation, confidence, enjoyment, happiness, and social inclusion in shaping children’s participation in physical activity and their overall health trajectory. Together, these conceptual developments illustrate a shift from viewing motor competence as a fixed set of physical skills toward understanding it as a dynamic and multifaceted construct, one that influences and is influenced by a wide range of behavioral, emotional, and contextual factors throughout childhood and beyond.

Several studies have recently examined the links between MC and various health-related aspects. MC is positively associated with higher levels of PA [6,20,21] and HRF [22,23,24]. In contrast, the inverse association between MC and WS has been confirmed [25,26,27]. For example, in a review by Holfelder & Schott [21], the authors analyzed cross-sectional studies predominantly involving children aged 6–12 years from diverse backgrounds and found strong evidence of a positive relationship between MC and organized PA. However, they also noted that MC competency had only low predictive value for PA levels in adulthood. Utesch et al. [24], in their review of 19 eligible studies, reported that the correlation between MC and HRF increased with age, showing moderate associations in early childhood and strong associations in early adulthood. The samples included both boys and girls, typically between the ages of 4 and 18, with varying levels of PA. Biino et al. [25] conducted a longitudinal study on 117 children (both boys and girls) aged 8–13 years over the course of 4 years. They found that the MC level and its development were closely tied to WS, with children of a higher WS showing delayed MC progression, emphasizing the bidirectional relationship between these constructs.

However, most of the above-mentioned studies investigating relationships between MC and PA as well as HRF and WS during middle childhood have been focused on exploring relationships between MC and one health-related variable separately (e.g., MC and PA, MC and HRF, or MC and WS) and then on the relationships of all these variables together [6]. By exploring relationships between MC and PA as well as HRF and WS together, the information obtained may provide a much more comprehensive view of the importance of MC concerning key health-related variables such as PA, HRF, and WS during the overall development of children. For example, Jaakkola et al. [28] explored the relationships between MC, perceived MC, HRF, and PA in primary school-age children from only fifth grade. In their cross-sectional study, the authors found a direct path from MC to PA only in boys and an indirect path from MC to PA through the mediation effect of HRF for both genders. However, the assessment of WS was not included, and three product-oriented tests for the evaluation of MC, with two of them having rather being characteristic of physical fitness tests, seemed to be the weakness of the study. In a four-year follow-up study by Rodrigues et al. [29], children with a greater increase in HRF and MC during childhood (from 1st to 4th grade in primary school) were found to be less prone to developing unhealthy WS in the form of overweight or obesity. However, the data about children’s PA were not explored in this study, and again, three product-oriented tests assessing agility, speed, and explosive power seemed to be tests of physical fitness rather than of the quality of movement patterns in the framework of children’s overall MC. To avoid such overlap in tests assessing the levels of MC and HRF, the choice of appropriate research tools should be crucial for the correct settings of the research’s methodological design [24].

Given the identified gaps, the present study aimed to examine the associations between motor competence (MC) and three key health-related variables—physical activity (PA), health-related fitness (HRF), and weight status (WS)—in a single, comprehensive model focused on children aged 9–10 years. This age range, representing middle childhood, was deliberately selected due to its developmental significance; it marks a period when FMSs stabilize and strongly influences children’s engagement in organized activities and sports. Importantly, studies often group early and middle childhood together, which may obscure age-specific relationships due to variability in maturation and motor development. We sought to reduce potential confounding and yield clearer insights by isolating this age group. Although this study did not include mediation analysis, we expanded the literature by jointly analyzing MC’s relationships with PA, HRF, and WS, thereby capturing a broader and more integrated picture of these interconnected variables. We hypothesized that higher levels of MC would be associated with greater PA and HRF and a healthier WS. We further expected that children with higher MC levels would display significantly more favorable health profiles than those with lower MC levels. Differences between the moderate and high MC groups were analyzed exploratorily, given the limited and mixed evidence in the previous literature.

## 2. Materials and Methods

### 2.1. Participants

Typical developing, middle-school-aged children (n = 86:59 boys and 27 girls; see Table 1) aged 9–10 years old (10.1 ± 0.6 years) from a primary school in the region of Central Bohemia in the Czech Republic participated in this cross-sectional observational study. The study aimed to assess the associations between motor competence and health-related variables at a single time point. The school was chosen based on its willingness to participate and its demographic representativeness of the regional population. All eligible children within the selected age range were invited to participate, making this a convenience sample rather than a randomized one.

The inclusion criteria required the children to be between 9 and 10 years of age and have written informed consent provided by a parent or legal guardian, along with child assent, in accordance with ethical standards. The exclusion criteria included any medical problems that compromised participation in the study. Children diagnosed with mental or other clinically diagnosed impairments (such as ADHD, DCD, or developmental dysphasia) and those with special needs were excluded. Additionally, children were not excluded based on their prior physical activity levels, but data regarding participation in organized sports were recorded. At the time of testing, 29% of the children had not participated in organized sports, while 71% had either participated in organized sports (e.g., football, floorball, volleyball, basketball, track and field, gymnastics, combat sports, firefighting, or swimming) for at least 2 years or had a minimum of 1 year of experience.

In cooperation with the school management, parents were informed of the study’s purpose, procedures, and benefits. The Faculty of Physical Education and Sport Ethics Committee at Charles University in Prague approved the study (No. 207/2016). All the measurements for this study were conducted from January to April 2022 and performed by academic and school staff trained in administering all tests.

### 2.2. Assessment of Motor Competence (MC)

The Test of Gross Motor Development, 2nd edition (TGMD-2) [30], designed for children aged 3–10 years, was used to assess the level of children’s MC proficiency. Although the TGMD-2 has not been standardized for the Czech population, it was selected for this study due to its strong international reputation as a valid and reliable tool for assessing fundamental motor skills in children aged 3–10 years [14,30]. The test has been applied in various countries with different cultural backgrounds, demonstrating adequate construct validity and reliability [13,31,32]. Furthermore, the aim of our study was to conduct relative comparisons within the sample rather than to benchmark performance against U.S.-based normative data. To support the reliability of the assessment in our local context, intra-rater agreement was evaluated using Cohen’s kappa, yielding excellent reliability (κ = 0.87). The TGMD-2 consists of 12 movement skills (6 locomotor and 6 object control) administered according to test guidelines [30]. Prior to performing each skill test, participants were given an explanation and a demonstration by the trained test examiner. Then, each child had two official attempts to perform the skill. The whole test procedure was video recorded and later evaluated by a single examiner. Raw scores, standard scores, and percentiles were calculated for the locomotor and object-control subtest and the overall TGMD-2. The motor quotient that allows descriptive ratings for the TGMD-2 was also calculated. We decided to use all mentioned values of the TGMD-2, as Bolger et al. [13], in their review, recommended that all future studies should report these values when the MC level—using the TGMD-2—is presented to allow comparisons across studies.

### 2.3. Assessments of Physical Activity (PA)

Weekly PA among the children was assessed using ActiGraph GT3X accelerometers (ActiGraph, LLC, Inc., Fort Walton Beach, FL, USA). Children were instructed to wear the devices on the right hip during waking hours and to remove them only for sleeping and water-based activities (e.g., showering or swimming) for seven consecutive days, always from Monday to Sunday. A written form of instruction was also provided for the children and parents. Class teachers were asked to monitor the wearing and correct placement of the devices while the children were at school. The parents were asked to monitor the wearing during the free time of the children. An Actigraph accelerometer was reported as a valid and reliable method for assessing PA among children [33]. For this study, the two variables, steps and moderate-to-vigorous PA (MVPA), were calculated and presented as mean daily values (steps (number)/day; MVPA (minutes)/day). A 1 s epoch and cutoff points for children’s PA were used [34], with MVPA defined as ≥2296 counts per minute. According to Denstel et al. [35], children were excluded from overall analyses if accelerometer data (waking wear time) were not obtained for at least 10 h per day on at least four weekdays and one weekend day (n = 16 with violation of these rules).

### 2.4. Assessments of Health-Related Fitness (HRF)

A Unifittest 6–60 test battery was used to measure the HRF components of the children. The test is standardized for the Czech population 6–60 years of age [36,37] with a satisfactory level of reliability and validity [36] and consists of four physical fitness parameters (shuttle run 4 × 10 m, standing broad jump, sit-ups, and 20 m progressive shuttle run). The shuttle run 4 × 10 m assesses coordination and speed, and it was performed twice by each participant, with 3–4 min of rest between the two trials. In a starting position, the participant stood on the starting line without moving into the space between photocells. The participant sprinted to the opposite marker (10 m), turned, and returned to the starting line directly adjacent to the photocell gate. This was performed twice to cover a 40 m distance. The time of the faster trial was recorded. An infrared timing gate (Alge Timing GmbH, Lustenau, Austria) placed at approximately hip height was used for the start and finish points. The standing broad jump, an indicator of explosive power in the lower limbs, was performed three times by each participant, with 2 min of rest between trials. The participant stood behind a line marked on the ground. A two-foot takeoff and landing area was used, and participants were instructed to jump as far as possible while swinging their arms and bending their knees to provide forward momentum. The longest jump was recorded and used for the analysis. A progressive shuttle run of 20 m is a measure of maximal aerobic fitness. The participant continuously ran between two lines 20 m apart, keeping pace with recorded beeps, which accelerated each minute. The test was stopped when the participant failed to reach the line (within two meters) after two consecutive warnings. Finally, from each test item, a standard score was obtained. The composite standard score of all tests on a scale from 0 to 20 was calculated as a marker of total HRF.

### 2.5. Assessments of Weight Status (WS)

WS was classified according to body mass index (BMI), which is commonly calculated as body weight (kg) divided by body height (meter squared). We used percentile cut-off points to define a group of children with normal weights and a group of children falling into the categories of overweight and obese, following the general recommendations of the World Health Organization [38]. A BMI beyond the 85th percentile resulted in labeling children as overweight, and a BMI beyond the 95th percentile resulted in labeling children as obese. A BMI ranging between the 25th and 84th percentile was used to determine children with normal weight. Using the described criteria, 70% (n 68) of the children with normal weights were labeled as children with a healthy WS, and 20% (n 18) of the overweight and obese participants were labeled as children with an unhealthy WS. A medical calibrated scale TPLZ1T46CLNDBI300 (Helago s.r.o., Přerov, Czech Repulic) assessed body weight to the nearest 0.1 kg. A P375 portable anthropometer (TRYSTOM Ltd., Olomouc, Czech Republic) was used to measure participants’ heights. Measurements were taken to the nearest 0.1 cm.

### 2.6. Data Collection Process

All tests were conducted from January to April 2022 during regular physical education lessons within the school timetable. Due to logistical constraints and varying child attendance, tests were not performed in a fixed sequential order. Instead, children were assessed in small rotating groups at different testing stations set up in multiple locations within the school. Each group progressed through the stations based on availability and supervision by trained staff. This flexible approach ensured all children were tested under consistent conditions while minimizing disruption to school routines. Despite the variable order, care was taken to standardize instructions and measurement protocols across all test stations. To avoid the impact of fatigue on performance, physically demanding tests—such as the 20 m progressive shuttle run and the 4 × 10 m shuttle run—were scheduled toward the end of each child’s testing rotation or appropriately spaced with low-demand assessments. This helped ensure that fatigue from high-intensity tasks did not negatively affect performance in subsequent tests.

To minimize potential order effects, the children completed the tests in a randomized rotational order across small groups, and the more physically demanding tests (e.g., the 20 m shuttle run and 4 × 10 m shuttle run) were consistently scheduled at the end or spaced apart by low-effort activities. This procedure helped to reduce fatigue and sequencing bias in the performance outcomes.

### 2.7. Statistical Analysis

All statistical analyses were performed using SPSS version 24.0 (IBM Corp., Armonk, NY, USA) with a significance level set at α = 0.05. The normality of the data distribution was assessed using the Shapiro–Wilk test. We used the Mann–Whitney U test to find possible gender differences in MC, HRF, PA, and WS with a subsequent assessment of the effect size using Hedges’ g, where the magnitudes are presented as follows: 0.3 (“small” effect), 0.5 (“medium” effect), and 0.8 (“large” effect). A chi-squared test was employed to investigate the relationships between categorical pairs of variables (MC-HRF, MC-PA, and MC-WS). For the chi-squared test, children were categorically divided into three groups of MC: high (children who achieved average or above average results on the TGMD-2), moderate (children who performed below average on the TGMD-2), and low (children who performed poorly or extremely poorly on the TGMD-2), HRF (“above average, average, and below average”, based on the verbal assessment according to the Unifittest 6–60 test standard), PA (“good, fair, and poor” as percentile tercile groups according to the results of the MVPA), and into two groups of WS (“healthy WS and unhealthy WS”, according to cut-off points of international standards for normal weight, obesity, and being overweight). Two 3 × 3 contingency tables and one 3 × 2 contingency table were constructed to analyze the association between MC as an independent variable (three MC groups: high, moderate, and low) and HRF (three HRF groups: above average, average, and below average), PA (three PA groups: good, fair, and poor) and WS (two WS groups: healthy and unhealthy) as dependent variables in our analysis. Adjusted standardized residuals were used to calculate the difference between the observed and expected counts and divided by the estimated standard error to determine the statistical significance of individual cells. Cells with adjusted standardized residuals that were sufficiently large (typically exceeding a critical value, often set at ±1.96 for a 95% confidence level) were considered statistically significant. Then, because of an increased risk of making a Type I error (false positives) due to the increased number of comparisons, a Bonferroni correction was made. This adjusted significance level was then used to determine each test’s statistical significance [39]. According to Cohen [40], Cramer’s V as the effect size was used to measure the strength of association in the chi-squared test, with interpretations of 0.1 (“small” effect), 0.3 (“medium” effect), and 0.5 (“large” effect). To compare the median scores of the HRF, PA, and WS between children with different levels of MC (three groups, “high, moderate, and low”, based on the same distribution principle as in the previous analysis), the Kruskal–Wallis test and Dunn’s post hoc test were used. The effect size’s partial eta squared (η^2^) was calculated to quantify the magnitude of differences observed in the Kruskal–Wallis test. According to Cohen [41], the values of η^2^ are interpreted as follows: 0.01 for a “small” effect, 0.06 for a “moderate” effect, and 0.14 for a “large” effect.

## 3. Results

The values of the baseline characteristics, individual tests, and their subtests for all children together, as well as girls and boys separately, are presented in Table 1. Age, body height, and weight were significantly higher among the boys (*p* < 0.05, g = 0.54–0.83) compared with the girls. There were no significant gender differences in the TGMD-2 or its subtests. Girls scored significantly better in the 4 × 10 m shuttle run test (ss) (*p* < 0.05, g = 0.55) compared with boys. However, we found no gender differences in the Unifittest 6–60 total scores.

The results of the chi-squared tests for associations between the MC and HRF levels, MC and PA levels, and MC and WS levels are presented in Table 2, Table 3 and Table 4. When analyzing the association between MC and HRF (Table 2), no cell was expected to return a count of less than five and χ^2^ (4, N = 86) = 29.835, where *p* = 0.00001 (*p* < 0.05) and Cramer’s V = 0.42, indicating a strong positive association between MC and HRF. In children with high MC levels, it was revealed that 59.3% performed above average and only 7.4% below average in terms of HRF. On the other hand, among the children with low MC levels, only 10% performed above average and 70% performed below average in terms of HRF. The second multiple 3 × 3 chi-squared test evaluated the association between MC and PA levels (Table 3). Again, no cell was expected to return a count of less than five, and the χ^2^ (4, N = 86) = 10.397, where *p* = 0.03424 (*p* < 0.05) and Cramer’s V = 0.25, revealing a moderate positive association between MC and PA. In children classified in the high MC group, 48.1% performed well in terms of PA. On the contrary, 53.3% of the children classified in the low MC group reached poor PA levels. In the third chi-squared analysis (Table 4), where we tested the relationship between MC and WS, the values of χ^2^ (2, N = 86) = 10.129, *p* = 0.00632 (*p* < 0.05), and Cramer‘s V = 0.34 again showed a moderate negative correlation. The analysis indicated that 88.9% of the high-MC group noted a healthy WS, and only 11.1% noted an unhealthy WS. Conversely, 40% of the low-MC group met the status of an unhealthy WS.

The results in Table 5 indicate significantly higher performance by children in the high-MC group in all measured variables of HRF, PA, and WS compared with the children with low MC levels. Similarly, children with moderate MC levels outperformed children with low MC levels in all measured variables of HRF and WS (except for PA variables). There were no significant differences in all variables between children with high or moderate MC levels.

## 4. Discussion

The current study delved into an examination of the relationships between motor competency (MC), physical activity (PA), healthy physical fitness (HRF), and weight status (WS) in Czech children during middle childhood (age 9–10 years), addressing the limited number of studies that have comprehensively examined these interrelationships in one sample. Additionally, it sought to uncover potential differences in HRF, PA, and WS among children with varying levels of MC. The findings of this study highlight a significant association between MC and HRF with a strong effect, a significant moderate association between MC and PA, and a moderate effect between MC and WS in children during middle childhood. Moreover, children with higher proficiency in MC demonstrated significantly higher levels of PA and HRF and a healthier WS than those with lower MC proficiency. Similarly, children with moderate MC outperformed children with low MC in terms of HRF and WS but not PA. There were no significant differences in all measured variables between children with high and moderate levels of MC. These results underscore the importance of promoting and enhancing MC during childhood as a means of fostering overall health and well-being, including HRF, PA, and WS.

As we assumed in our first hypothesis, we found significant moderate-to-strong relationships between MC and PA as well as HRF and WS. These findings confirm Stodden’s theoretical model [10], according to which the relationships between MC and other health variables become stronger during middle (9–10 years) and late childhood (11–12 years) compared with early (3–5 years) and early middle (6–8 years) childhood, where no or low relationships are present [6,13]. Our findings of a moderate association between MC and PA in children aged 9–10 years align with the theoretical assumptions proposed by Stodden et al. [10], who suggested that the relationship between MC and PA strengthens with age. However, as our study did not include comparisons across age groups, we cannot empirically support this developmental progression. Future research should examine MC–PA relationships longitudinally to validate age-related differences proposed in developmental models. As children transition from early to later stages of childhood, the cumulative influence of individual characteristics and environmental factors increasingly shapes both their MC and PA behaviors [18]. Higher levels of MC competence acquired during middle and later childhood provide individuals with a broader motor repertoire, enhancing their ability to participate and excel in diverse physical activities, sports, and games throughout their lives [10,18]. Moreover, this idea is supported by theories such as the dynamic systems theory [42] and the ecological systems theory [43]. These theories emphasize how motor development emerges through complex interactions between biological, environmental, and task-related constraints. The observed differences in PA, HRF, and WS based on MC levels align with the premise that higher MC proficiency enables children to adapt more effectively to diverse movement contexts. This supports the notion that motor competence is not only a product of maturation but also an adaptable capacity influenced by learning opportunities, consistent with the ecological approach. Thus, our findings reflect how MC acts as a key factor within these theoretical frameworks, mediating the interaction between personal abilities and environmental challenges.

The strong positive association between MC and HRF found in our study corresponds with the results of several key comprehensive reviews [21,24,44], which similarly confirmed a moderate-to-strong positive association between MC and HRF that strengthened with increasing age. Furthermore, den Uil et al. [6] measured the association between MC and HRF in children aged from 6 to 14 years and found that this association was strongest (r = 0.6) in 9- and 10-year-old children. On the other hand, some studies, such as that by Wu et al. [45], found only a weak relationship between FMSs and HRF in children from first to fourth grade of elementary school, suggesting that the FMS level could not predict the level of HRF. We think one possible cause for this weak relationship is the grouping of early-middle and middle childhood values together, where the presence of the scores from the younger children may decrease the chances of stronger associations with those of older children. This was one reason why we focused our attention exclusively on middle childhood children, namely to reduce potential bias by including a wide range of children from different age groups. Often, overlap in content between measures of MC proficiency and HRF has been mentioned as another reason for the association strength bias. Therefore, the choice of appropriate research tools should be properly considered [24]. For this reason, we used a process-oriented test to assess the quality of movement skills and, conversely, a product-oriented test to assess select physical abilities in our study.

We also revealed a significant moderate inverse association between MC and WS. Similar findings have been reported in several review papers focused on children and adolescents [23,46,47] and in comprehensive studies which included more health-related variables [6,15,48]. Similarly, while verifying associations in Stodden’s conceptual model, den Uil et al. [6] found a significant association between MC and WS only from the age of 9 years. As a higher WS was found to reduce MC proficiency in children significantly [27], WS has been marked as an important correlate of MC proficiency in children [15]. On the other hand, Slotte et al. [49] suggested that the association between MC and WS is still unclear because of the interchangeable use of the “motor skills” and “motor abilities” test items within the MC test batteries.

Regarding our second hypothesis, we found that during middle childhood, children with higher levels of MC achieved significantly higher levels of PA and HRF and a lower WS compared with children with lower MC levels. These findings unequivocally confirm the conceptual model of Stodden et al. [10] and the model’s associations during middle childhood. As the practical significance of differences between children with high and low MC proficiency reached large effects (HRF and its separated test items) and a medium effect (PA and WS), one could consider the existence of Seefeldt’s theory of a proficiency barrier [19] and the metaphor of the mountain of motor development [18]. Based on these approaches, children with high (sufficient) levels of MC will stay active and successfully transfer MC into lifetime physical activities and context-specific motor skills, but children with low MC levels will be less successful and ultimately drop out of physical activities at higher rates. In our opinion, the important question raised in our study is the following: What is the required level of MC proficiency that ensures positive spiral engagement in Stodden’s model? In other words, what is the minimum necessary level of MC based on a used diagnostic tool which still ensures sufficient levels of PA and HRF and a healthy WS? The overall level of MC found in our study was low (20.7 ± 19.2 percentiles on the TGMD-2 for all children), which corresponds with the previous findings of Bolger’s review [13], focused on searching for the global level of FMSs. Our study’s group with high MC included children who scored from the 27th to 84th percentile on the total TGMD-2. Such a range of MC proficiency seems to be too wide, and thus it remains unclear which concrete value or range of values of the total score in adequately chosen MC proficiency diagnostic tools could be sufficient for the healthy development of children, especially in middle childhood, where most associations between MC and other health variables begin to be significant or stronger.

We consider the comprehensive exploration of the importance of MC with respect to all substantial health-related variables (PA, HRF, and WS) during a narrow age period of middle childhood (9–10 years) to be the main strength of this study. Many studies that explored the associations between MC and HRF used physical fitness test items within the evaluation of MC proficiency—which led to probable bias in the strength of associations—and thus we decided to use suitable assessment tools (TGMD-2 for assessment of movement quality and Unifittest 6–60 for assessment of physiological base of motor abilities) that did not overlap in their test items. This study has several limitations. First, the sample was relatively small and drawn from a single school, which may limit the generalizability of the findings. Second, there was no randomization process; children were selected based on school participation and availability. Third, although gender comparisons were explored descriptively, the sample included almost twice as many boys as girls, potentially skewing group-based outcomes. Fourth, while MC levels were categorized into three proficiency groups, the score distributions were not evenly spaced across the TGMD-2 percentile scale, limiting precision in defining the thresholds for “high”, “moderate”, or “low” competence. Finally, the cross-sectional nature of the study precludes any causal inference and does not account for seasonal effects or changes over time. This design also cannot account for natural developmental variability, such as the attainment of motor milestones at different ages due to biological maturation. Therefore, we cannot determine whether low MC at ages 9–10 precludes improvement or engagement at later stages, such as age 12. Future studies should incorporate longitudinal designs, stratified gender analysis, and a more representative, randomized sample.

## 5. Conclusions

This study identified significant associations between motor competence and physical activity, health-related fitness, and weight status in Czech children aged 9–10 years. These findings suggest that children with higher motor competence tend to demonstrate more favorable health-related characteristics. The results reinforce the growing evidence that motor competence is a meaningful factor in children’s health and well-being during middle childhood. While the cross-sectional nature of this study prevents us from drawing causal conclusions, the observed associations highlight a compelling rationale for further research. These findings may inform the development of well-designed longitudinal studies and targeted interventions aimed at improving motor skills as part of comprehensive health promotion strategies. Natural variation in neural and physical development should also be considered in interpreting differences in MC levels at this age. Although caution should be exercised when translating these results into policy recommendations, this study provides valuable insights that contribute to the understanding of how motor competence relates to multiple health indicators. It underscores the potential of early support for motor development within school and community settings as part of a broader approach to fostering children’s physical literacy and lifelong engagement in healthy behaviors.

## Figures and Tables

**Table 1 jfmk-10-00258-t001:** Anthropometric characteristics and MC, HRF, PA, and WS performance.

	All Children	Boys	Girls	Hedges’ g
Age (years)	10.1 ± 0.6	10.3 ± 0.6 **	9.8 ± 0.6 **	0.83 ^⸸⸸^
Hight (cm)	142.2 ± 7.0	143.4 ± 7.2 *	139.7 ± 6.0 *	0.54 ^⸸^
Weight (kg)	37.0 ± 9.6	38.9 ± 10.6 *	33.0 ± 5.1 *	0.64 ^⸸^
TGMD-2 (mq)	84.5 ± 19.2	84.0 ± 12.8	85.6 ± 9.8	0.13
TGMD-2 (pct)	20.7 ± 19.2	20.9 ± 20.1	20.2 ± 17.5	0.04
TGMD-2 (ss)	14.8 ± 4.0	14.7 ± 4.3	15.2 ± 3.3	0.12
Locomotor subtest (pct)	34.9 ± 24.1	34.6 ± 23.5	35.5 ± 25.7	0.04
Locomotor subtest (ss)	8.5 ± 2.4	8.5 ± 2.3	8.6 ± 2.5	0.04
Object-control subtest (pct)	16.8 ± 18.0	17.5 ± 19.4	15.5 ± 14.7	0.11
Object-control subtest (ss)	6.3 ± 2.4	6.2 ± 2.7	6.6 ± 1.6	0.17
UNIFITTEST 6–60 (ss)	20.9 ± 7.2	20.3 ± 8.0	22.1 ± 5.1	0.25
Standing broad jump (ss)	4.8 ± 2.3	4.8 ± 2.6	4.9 ± 1.7	0.04
Sit-ups (ss)	6.9 ± 2.0	6.7 ± 2.0	7.3 ± 1.7	0.31
Shuttle run 4 × 10 m (ss)	5.5 ± 2.3	5.1 ± 2.3 *	6.3 ± 1.9 *	0.55 ^⸸^
20 m progressive shuttle run (ss)	3.7 ± 2.0	3.8 ± 2.2	3.6 ± 1.3	0.10
MVPA (min)	69.2 ± 38.2	69.3 ± 40.8	68.9 ± 32.6	0.01
Steps	9561.9 ± 3512.5	9534.4 ± 3801.2	9622.2 ± 2845.5	0.03
BMI (kg/m^2^)	18.1 ± 3.3	18.7 ± 3.7	16.8 ± 1.8	0.59 ^⸸^

* *p* < 0.05. ** *p* < 0.01. ⸸ Medium effect size. ⸸⸸ Large effect size. Abbreviations: mq = motor quotient; pct = percentile; ss = standard score; TGMD-2 = Test of Gross Motor Development, 2nd edition; MVPA = moderate-to-vigorous physical activity; BMI = body mass index.

**Table 2 jfmk-10-00258-t002:** Chi-square cross-table indexes for MC levels × HRF levels.

			HRF Levels			Total
			Above Average	Average	Below Average	
MC levels	High	Count	16	9	2	27
		Expected Count	9.1	8.5	9.4	27
		% within	59.3%	33.3%	7.4%	100%
		Adj. St. Res.	3.4 *	0.3	−3.6 *	
	Moderate	Count	10	12	7	29
		Expected Count	9.78	9.11	10.12	29
		% within	34.5%	41.4%	24.1%	100%
		Adj. St. Res.	0.1	1.4	−1.5	
	Low	Count	3	6	21	30
		Expected Count	10.1	9.4	10.5	30
		% within	10.0%	20.0%	70.0%	100%
		Adj. St. Res.	−3.4 *	−1.7	5.0 *	
Total		Count	29	27	30	86
		Expected Count	29.0	27.0	30.0	86
		% within	33.7%	31.4%	34.9%	100%

Adj. Std. Res. = adjusted standard residuals. * Statistically significant difference between observed and expected counts.

**Table 3 jfmk-10-00258-t003:** Chi-squared cross-table indexes for MC levels × PA levels.

			PA Levels			Total
			Good	Fair	Poor	
MC levels	High	Count	13	8	6	27
		Expected Count	8.8	9.1	9.1	27
		% within	48.1%	29.6%	22.2%	100%
		Adj. St. Res.	2.1	−0.5	−1.5	
	Moderate	Count	9	13	7	29
		Expected Count	9.4	9.8	9.8	29
		% within	31.0%	44.8%	24.1%	100%
		Adj. St. Res.	−0.2	1.6	−1.3	
	Low	Count	6	8	16	30
		Expected Count	9.8	10.1	10.1	30
		% within	20.0%	26.7%	53.3%	100%
		Adj. St. Res.	−1.8	−1.0	2.8 *	
Total		Count	28	29	29	86
		Expected Count	28.0	29.0	29.0	86
		% within	32.6%	33.7%	33.7%	100%

Adj. Std. Res. = adjusted standard residuals. * Statistically significant difference between observed and expected counts.

**Table 4 jfmk-10-00258-t004:** Chi-squared cross-table indexes for MC levels × WS levels.

			WS Levels		Total
			Healthy	Unhealthy	
MC levels	High	Count	24	3	27
		Expected Count	21.3	5.7	27.0
		% within	88.9%	11.1%	100.0%
		Adj. St. Res.	1.5	−1.5	
	Moderate	Count	26	3	29
		Expected Count	22.9	6.1	29.0
		% within	89.7%	10.3%	100.0%
		Adj. St. Res.	1.7	−1.7	
	Low	Count	18	12	30
		Expected Count	23.7	6.3	30.0
		% within	60.0%	40.0%	100.0%
		Adj. St. Res.	−3.2 *	3.2 *	
Total		Count	68	18	86
		Expected Count	68.0	18.0	86.0
		% within	79.1%	20.9%	100.0%

Adj. Std. Res. = adjusted standard residuals. * Statistically significant difference between observed and expected counts.

**Table 5 jfmk-10-00258-t005:** Statistics for the comparison between MC groups’ HRF, PA and WS variables.

	MC Levels					
	High	Moderate	Low	Test Statistic	*p* Value	η^2^
Unifittest 6–60 (ss)	26.0 ± 4.6 ^a^	22.0 ± 5.8 ^c^	15.1 ± 6.3 ^a,c^	33.6	*p* < 0.01	0.38 ^⸸⸸^
Standing broad jump (ss)	6.3 ± 1.7 ^a^	5.1 ± 2.1 ^b^	3.3 ± 2.1 ^a,b^	22.9	*p* < 0.01	0.25 ^⸸⸸^
Sit-ups (ss)	7.9 ± 1.5 ^a^	7.2 ± 1.8 ^b^	5.7 ± 1.9 ^a,b^	18.7	*p* < 0.01	0.20 ^⸸⸸^
Shuttle run 4 × 10 m (ss)	6.8 ± 1.5 ^a^	5.9 ± 2.1 ^c^	3.8 ± 2.0 ^a,c^	26.1	*p* < 0.01	0.29 ^⸸⸸^
20 m progressive shuttle run (ss)	5.1 ± 1.9 ^a^	3.8 ± 1.5 ^c^	2.4 ± 1.6 ^a,c^	28.9	*p* < 0.01	0.32 ^⸸⸸^
MVPA (min/day)	87.1 ± 46.4 ^b^	68.9 ± 33.9	53.2 ± 26.1 ^b^	7.6	*p* < 0.05	0.07 ^⸸^
Steps (number/day)	11,012.4 ± 4246.6 ^b^	9598.6 ± 3032.1	8221.1 ± 2701.5 ^b^	8.3	*p* < 0.05	0.08 ^⸸^
BMI (kg/m^2^)	17.3 ± 1.6 ^b^	17.4 ± 3.5 ^d^	19.6 ± 3.8 ^b,d^	9.4	*p* < 0.01	0.09 ^⸸^

Values are mean ± SD. b,c *p* < 0.05. a,d *p* < 0.01. ⸸ Medium effect size. ⸸⸸ Large effect size. Abbreviations: ss = standard score; MVPA = moderate-to-vigorous physical activity; BMI = body mass index.

## Data Availability

The datasets used in this study are available from the corresponding author upon reasonable request due to privacy and ethical reasons.

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
