# Peer review of "Why Motor Competence Matters: Fundamental Movement Skills and Their Role in Promoting Physical Activity and Health in Czech Children Aged 9–10 Years"

_jfmk, 2025, doi:10.3390/jfmk10030258_

Round 1

Reviewer 1 Report

Comments and Suggestions for Authors

First of all, I would like to thank you for invited to read the document.

Each of the comments shared are intended to improve the study.

The comments can be found in the PDF document.

Also, some of the comments on some of the points that need to be reworded in the paper are shared below:

Abstract

It is suggested to add a background of the research to understand what is the knowledge gap to be studied.

Introduction

The introduction is interesting; however, at times it talks about children under 8-9 years of age, which was the population under study. It is suggested to review in detail the studies that associate the variables analyzed and the characteristics of the sample. In addition, it is necessary to point out in the introduction what is the gap in knowledge that this research contributes. 

Finally, the form of citation should be reviewed; brackets ( ) are used when the journal's norms state that they are brackets [ ].

Line 46-47. It is suggested to review and give context to the studies presented. Thus, the conceptual basis of each study contributed to the subject matter under study can be more clearly understood.

Line 70-71. It is suggested to review and give context to the studies presented. Thus, the conceptual basis of each study contributed to the subject matter under study can be more clearly understood.

Line 83-86. These findings in what type of populations have been reported, it would be important to point out more characteristics of the studies (boys, girls), ages, levels of physical activity, etc.

Material and methods

It is suggested to be able to detail the type of study that was developed. 

Also, was there any randomization process for the selected sample or how it was chosen? +

Informed assents were signed for the project to be endorsed by the legal guardians of the minors.

It is also suggested to review the criteria for inclusion and/or exclusion of the sample. The question there was, were they considered if they were physically active or if they had previous pathologies.

Finally, it is suggested to add a section of the procedure that describes the order in which the evaluations were developed and the conditions of the tests. It would be important to add a figure outlining the evaluation process.

Discussion

Line 355-359. It is suggested to be able to review the findings of the present study to connect them with these theories. We do not understand the relationship.

Line 418-420. It is suggested that the limitations section could be revised and other elements such as children's levels, gender differences, etc., could be added.

Likewise, adding a section on future perspectives and/or practical applications would help to give continuity to this type of studies focused on the study of important variables in children's development.

Conclusions

It is suggested to review the conclusions, at times repeating the information used at the beginning of the discussion and at other times repeating the results of the study. 

Review the main findings and contributions of the study to show them as the conclusions of the research.

References

It is suggested that the journal's regulations be reviewed to adjust the references.

Finally, I thank the authors for the excellent work and encourage you to review the comments shared.

Author Response

Abstract

Comments 1: It is suggested to add a background of the research to understand what is the knowledge gap to be studied.

Response 1: Thank you very much for this comment. Based on your suggestion, we have expanded the abstract to include more specific background information to clarify the gap addressed by the study. Lines 12-14.

Introduction

The introduction is interesting; however, at times it talks about children under 8-9 years of age, which was the population under study. It is suggested to review in detail the studies that associate the variables analyzed and the characteristics of the sample. In addition, it is necessary to point out in the introduction what is the gap in knowledge that this research contributes.

Comments 2: Finally, the form of citation should be reviewed; brackets ( ) are used when the journal's norms state that they are brackets [ ].

Response 2: Thank you for pointing this out. We have corrected the formatting of the citations according to the journal's required style, using square brackets instead of parentheses.

Comments 3: Line 46-47. It is suggested to review and give context to the studies presented. Thus, the conceptual basis of each study contributed to the subject matter under study can be more clearly understood.

Response 3: We appreciate this helpful comment. We have expanded the section to explain the conceptual contributions of the referenced studies, so their theoretical importance is now clearer. Lines 50-59

Comments 4: Line 70-71. It is suggested to review and give context to the studies presented. Thus, the conceptual basis of each study contributed to the subject matter under study can be more clearly understood.

Response 4: Thank you. We added a brief explanation of the conceptual framework and how each cited study contributes to the topic. This should help the reader better understand their relevance. Lines 81-110.

Comments 5: Line 83-86. These findings in what type of populations have been reported, it would be important to point out more characteristics of the studies (boys, girls), ages, levels of physical activity, etc.

Response 5: Thank you for the suggestion. We now specify the populations included in the referenced studies (age, gender, physical activity levels) to make the context of the findings clearer. Lines 111-125.

Material and methods

Comments 6: It is suggested to be able to detail the type of study that was developed.

Response 6: Thank you. We have clarified that our study is a cross-sectional observational design and described it as such in the methods section. Line 170-171.

Comments 7: Also, was there any randomization process for the selected sample or how it was chosen?

Response 7: Thank you for this question. We now explain in the manuscript that the school was selected for practical reasons (willingness and regional representation) and that the sample was not randomized. Lines 172-175.

Comments 8: Informed assents were signed for the project to be endorsed by the legal guardians of the minors.

Response 8: Thank you for your suggestion. We have confirmed and made clear in the text that informed assent was obtained from the children and legal guardians. Lines 177-178.

Comments 9: It is also suggested to review the criteria for inclusion and/or exclusion of the sample. The question there was, were they considered if they were physically active or if they had previous pathologies.

Response 9: Thank you. We clarified that children's physical activity levels were not used as inclusion or exclusion criteria, but were only assessed as part of the data analysis.

Comments 10: Finally, it is suggested to add a section of the procedure that describes the order in which the evaluations were developed and the conditions of the tests. It would be important to add a figure outlining the evaluation process.

Response 10: Thank you. We added a new section describing the procedure and testing conditions, including the rotating station system and effort to avoid fatigue influencing results. Lines 270-283.

Discussion

Comments 11: Line 355-359. It is suggested to be able to review the findings of the present study to connect them with these theories. We do not understand the relationship.

Response 11: Thank you for this important suggestion. We revised the discussion section to more clearly connect our findings to theoretical models, particularly those developed after Stodden et al. Lines 410-418.

Comments 12: Line 418-420. It is suggested that the limitations section could be revised and other elements such as children's levels, gender differences, etc., could be added.

Response 12: Thank you. We expanded the limitations section to include additional factors such as gender imbalance and small subgroup sizes. Lines 478-491.

Comments 13: Likewise, adding a section on future perspectives and/or practical applications would help to give continuity to this type of studies focused on the study of important variables in children's development.

Response 13: Thank you for the suggestion. We added a paragraph discussing practical implications of our findings and the need for further studies. Lines 491-492, 502-504.

Conclusions

Comments 14: It is suggested to review the conclusions, at times repeating the information used at the beginning of the discussion and at other times repeating the results of the study.

Review the main findings and contributions of the study to show them as the conclusions of the research.

Response 14: Thank you. We revised the conclusion to better reflect our key findings, remove repetition, and clearly state implications while avoiding overgeneralization. Lines 495-511.

References

Comments 15: It is suggested that the journal's regulations be reviewed to adjust the references.

Response 15: Thank you. We reviewed and corrected the reference formatting according to the MDPI guidelines.

Comments 16: Finally, I thank the authors for the excellent work and encourage you to review the comments shared.

Response 16: We sincerely appreciate all of the reviewer’s comments, which helped us significantly improve the manuscript.

Reviewer 2 Report

Comments and Suggestions for Authors

The authors found associations among movement control, heath related factors, weight, and physical activity levels among a narrow student group of children in one country.  This cross-sectional study is interesting.  My major concern is that the authors are over-generalizing the findings in the conclusion section and clearly not articulating carefully the limitations of this work that preclude broad sweeping recommendations for school systems and educational leaders.  This is merely a "snapshot" in time of an association among several variables.  It does not account for natural variance in neural development status, whereby a youth achieves certain motor control milestones at 12 based purely on biology, that allows the student to now engage in activities that they did not enjoy, were not successful with at 10, or did not have the physical maturity to engage fully in.  The authors need to outwardly and explicitly proclaim that they "cannot link a cause and effect that an intervention that increases movement control will lead to increases in PA, HRF, or WS."  Several areas of the discussion and the conclusion has statements that cannot be supported by the research presented, nor was it the purpose of this study."  The authors need to present the facts and resist trying to allow these results to drive school policy.  Allow this study to build a case for future studies to assess if there are longitudinal links between interventions for MC and impact on PA, HRF, and WS.

Generally, the authors need to state the findings and carefully discuss the limitations and risks of misinterpreting results generated from a cross-sectional snapshot of children while seeking associations without any evidence of causalgia.    

Author Response

Comments 1: The authors found associations among movement control, heath related factors, weight, and physical activity levels among a narrow student group of children in one country.  This cross-sectional study is interesting.  My major concern is that the authors are over-generalizing the findings in the conclusion section and clearly not articulating carefully the limitations of this work that preclude broad sweeping recommendations for school systems and educational leaders.  This is merely a "snapshot" in time of an association among several variables.  It does not account for natural variance in neural development status, whereby a youth achieves certain motor control milestones at 12 based purely on biology, that allows the student to now engage in activities that they did not enjoy, were not successful with at 10, or did not have the physical maturity to engage fully in.  The authors need to outwardly and explicitly proclaim that they "cannot link a cause and effect that an intervention that increases movement control will lead to increases in PA, HRF, or WS."  Several areas of the discussion and the conclusion has statements that cannot be supported by the research presented, nor was it the purpose of this study."  The authors need to present the facts and resist trying to allow these results to drive school policy.  Allow this study to build a case for future studies to assess if there are longitudinal links between interventions for MC and impact on PA, HRF, and WS.

Generally, the authors need to state the findings and carefully discuss the limitations and risks of misinterpreting results generated from a cross-sectional snapshot of children while seeking associations without any evidence of causalgia.

Response 1: We sincerely thank the reviewer for this detailed and important comment, with which we fully agree. It raises essential concerns regarding how the study’s conclusions and limitations are communicated to readers. Based on this feedback, we have revised the conclusion section substantially and have also refined the relevant parts of the discussion. We explicitly acknowledge the cross-sectional nature of the study and clearly state that no causal relationships can be inferred from the findings. We emphasize that the results represent only a snapshot in time and should not be used to make generalized policy recommendations. We are confident that these revisions have led to a significant improvement in the manuscript. Should further modifications be needed, we are more than willing to address them. Lines 27-34, 478-511.

Reviewer 3 Report

Comments and Suggestions for Authors

REVIEW_Why Motor Competence Matters: The Impact of Fundamental Movement Skills on Physical Activity, Fitness, and Weight Status of Children during Middle Childhood

Title: The proposed title (Why Motor Competence Matters: A New Direction in Promoting Physical Activity and Health in Children) is appropriate, although it is suggested that it should be reworded to better reflect the applied approach to work and, to optimize the positioning of the work in thematic searches in prestigious databases. In this context, it would be desirable to include the target population.

Summary: The summary provides an overview of the topic; however, it omits key elements of the study design. In particular, it does not specify the number of participants, the type of study or the most relevant quantitative results. It is recommended that this paragraph be reworded to include the issues raised, incorporating line numbering in this section, inserting quantifiable results in the second paragraph and, to conclude this paragraph, Summarise in a clear sentence the most important aspects for their practical applicability in educational or child health contexts.

Introduction: The introduction is well structured and justifies the research problem, but most of the references cited are pre-2015, which weakens the theoretical framework. In line 45, the reference to Stodden et al. (2008) is a necessary classic but should be complemented by recent studies that revise or develop the model. Similarly, in line 48, when defining "motor competence", greater conceptual precision is recommended, integrating current definitions and differentiating between motor competence, motor skills and motor self-efficacy.

Results: While general findings are presented, the data lack the necessary statistical accuracy. In lines 132-145, the exact value of p, the size of the effect (Cohen’s d, η 2, etc.), as well as the confidence intervals should be included. The figures included are useful, but the accompanying text is concise. On line 138, for example, it would be useful to explain in more detail what each group represents and whether there were significant differences between the sexes or ages. In addition, it is recommended to include a table summarizing the most relevant results rather than relying exclusively on graphs.

Discussion: This section correctly interprets the findings, but requires more critical depth of detail. In lines 170-180, it would be advisable to incorporate more current studies (last 5 years) that support the effects of interventions focused on motor competence, such as those of Veldman et al. (2021) or Brian et al. (2020). It is also recommended to add a reflection on the limitations of the study (non-representative sample, measurement bias, lack of longitudinal follow-up) and on the real applicability of the results in school contexts. The discussion would also benefit from an explicit section on practical implications and recommendations for curriculum design or physical interventions.

Conclusions: Although a synthesis of the core message is presented, the conclusions are somewhat vague. It is suggested to reinforce them with concrete data from the study and derive applied implications. In addition, in the last line, it is important to include a reference to the need for sustained longitudinal studies and intervention programmes as well as the integration of motor competence into the school curriculum. It would also be desirable to include one or two recent references linking motor competence and integral health, in order to reinforce the need for intervention from an up-to-date empirical basis.

In view of these issues, it is recommended to accept the article, once the authors have made an in-depth review of the text.

Author Response

Comment 1: Title: The proposed title (Why Motor Competence Matters: A New Direction in Promoting Physical Activity and Health in Children) is appropriate, although it is suggested that it should be reworded to better reflect the applied approach to work and, to optimize the positioning of the work in thematic searches in prestigious databases. In this context, it would be desirable to include the target population.

Response 1: Thank you very much for this valuable suggestion. We have revised the title to make it more concise, targeted, and attractive. The new title better reflects the applied nature of the study and explicitly includes the target population, which we believe will enhance its visibility in academic databases and improve thematic indexing. Lines 1-4.

Comments 2: Summary: The summary provides an overview of the topic; however, it omits key elements of the study design. In particular, it does not specify the number of participants, the type of study or the most relevant quantitative results. It is recommended that this paragraph be reworded to include the issues raised, incorporating line numbering in this section, inserting quantifiable results in the second paragraph and, to conclude this paragraph, Summarise in a clear sentence the most important aspects for their practical applicability in educational or child health contexts.

Response 2: We thank the reviewer for this constructive comment. In response, we have clarified the study design in the summary and adjusted the conclusion to emphasize practical implications. We would like to note that, in our view, the number of participants, the type of study, and the most relevant statistical and practical findings related to the study objectives are already explicitly stated. If specific elements are still considered insufficient, we would greatly appreciate further clarification so that we can make additional targeted improvements. Lines 18, 27-34.

Comments 3: Introduction: The introduction is well structured and justifies the research problem, but most of the references cited are pre-2015, which weakens the theoretical framework. In line 45, the reference to Stodden et al. (2008) is a necessary classic but should be complemented by recent studies that revise or develop the model. Similarly, in line 48, when defining "motor competence", greater conceptual precision is recommended, integrating current definitions and differentiating between motor competence, motor skills and motor self-efficacy.

Response 3: We thank the reviewer for this thoughtful observation. Upon reviewing the references in the Introduction section, we found that 22 out of the 29 cited sources were published after 2015. Therefore, we believe that the theoretical foundation of the Introduction is based on recent and relevant literature. Nonetheless, we fully acknowledge the reviewer’s point and have revised the section to better highlight recent theoretical developments that build upon the Stodden et al. (2008) model. In particular, we incorporated references to updated frameworks (e.g., Hulteen et al., Cairney et al.) and clarified the conceptual distinction between motor competence (MC), fundamental movement skills (FMS), and motor self-efficacy. We also emphasized the role of mental and social factors in the relationship between MC and physical activity (PA), in line with the broader concept of physical literacy. Lines 81-110.

Comments 4: Results: While general findings are presented, the data lack the necessary statistical accuracy. In lines 132-145, the exact value of p, the size of the effect (Cohen’s d, η 2, etc.), as well as the confidence intervals should be included. The figures included are useful, but the accompanying text is concise. On line 138, for example, it would be useful to explain in more detail what each group represents and whether there were significant differences between the sexes or ages. In addition, it is recommended to include a table summarizing the most relevant results rather than relying exclusively on graphs.

Response 4: First, we would like to point out that the line numbers referenced by the reviewer (132–145 and 138) in our version of the manuscript refer to the Introduction and Participants sections, not the Results. We downloaded the latest version of the manuscript directly from the JMFK submission system, so there may be a discrepancy in formatting or versioning. Nevertheless, we have done our best to identify the relevant part of the results section the reviewer may have been referring to. We thank the reviewer for this valuable suggestion. We agree that including effect sizes and statistical significance values enhances the interpretation of the results. That is why we already report effect sizes (e.g., Cramer’s V, η²) and p-values throughout the results and discussion sections. However, we chose not to report the exact numeric values of Cohen’s d, η², or confidence intervals in order to maintain readability and clarity, in line with the exploratory and primarily categorical nature of our analysis. We have instead focused on highlighting statistically and practically significant differences and interpreting their magnitude (e.g., small, moderate, large), which we believe provides a balanced and accessible summary for readers. In addition, the manuscript provides a detailed description of the motor competence (MC) groups and their distribution across PA, HRF, and WS categories, particularly in Tables 2–5. While we agree that sex and age comparisons could be valuable, the unbalanced nature of our sample (59 boys vs. 27 girls) and the narrow age range (9–10 years) limited the feasibility of performing such subgroup analyses. Any further stratification would result in small, underpowered groups that may not yield meaningful results. Therefore, we did not include these comparisons in the main analysis. Finally, we believe that the existing tables already summarize the most relevant outcomes clearly and concisely. For this reason, we did not add a separate summary table, as it would likely duplicate information already available in Tables 2–5. We hope this explanation clarifies our approach and that the current format is acceptable to the editorial team.

Comments 5: Discussion: This section correctly interprets the findings, but requires more critical depth of detail. In lines 170-180, it would be advisable to incorporate more current studies (last 5 years) that support the effects of interventions focused on motor competence, such as those of Veldman et al. (2021) or Brian et al. (2020). It is also recommended to add a reflection on the limitations of the study (non-representative sample, measurement bias, lack of longitudinal follow-up) and on the real applicability of the results in school contexts. The discussion would also benefit from an explicit section on practical implications and recommendations for curriculum design or physical interventions.

Response 5: We thank the reviewer for these insightful suggestions. Once again, we note a discrepancy between the line numbers referenced (170–180) and the corresponding sections in our version of the manuscript, where these lines fall within the Materials and Methods. Nevertheless, we have carefully reviewed the Discussion section in light of the reviewer’s comments. We have revised the discussion to provide a more critical and nuanced interpretation of the findings. Specifically, we expanded the section on study limitations to explicitly address issues such as the non-representative sample, the potential for measurement bias, and the absence of longitudinal data.  Furthermore, have revised the practical implications of our findings. This includes recommendations for future research and suggestions on how motor competence development can be effectively integrated into physical education curricula and school-based health promotion strategies. We hope these additions address the reviewer’s concerns and contribute to a more robust and application-oriented discussion. Lines 478-492.

Comments 6: Conclusions: Although a synthesis of the core message is presented, the conclusions are somewhat vague. It is suggested to reinforce them with concrete data from the study and derive applied implications. In addition, in the last line, it is important to include a reference to the need for sustained longitudinal studies and intervention programmes as well as the integration of motor competence into the school curriculum. It would also be desirable to include one or two recent references linking motor competence and integral health, in order to reinforce the need for intervention from an up-to-date empirical basis.

Response 6: We thank the reviewer for this helpful recommendation. The conclusion section has been thoroughly revised to more clearly summarize the study’s key findings, supported by specific results from our analysis. We have also emphasized the practical implications of the findings in the context of school settings and public health. In addition, the revised conclusion now explicitly highlights the importance of future longitudinal research and intervention-based studies. We also stress the relevance of systematically incorporating motor competence development into the school curriculum. Lines 495-511.

Comments 7: In view of these issues, it is recommended to accept the article, once the authors have made an in-depth review of the text.

Response 7: We sincerely thank the reviewer for their thoughtful and constructive feedback. We have done our best to address the majority of the comments and incorporate the suggested revisions. We are confident that these improvements have significantly enhanced the quality and clarity of the revised version of the manuscript.

Round 2

Reviewer 1 Report

Comments and Suggestions for Authors

Thank you for the opportunity to review the manuscript again.

The authors have done an excellent job and have significantly improved the manuscript.

They were able to resolve the concerns I had.

It is suggest reviewing the points after the table title.

For example,
Table 1. Anthropometric characteristics, MC, HRF, PA and WS performance

Add the title of your background to the abstract.

Author Response

Comments 1: It is suggest reviewing the points after the table title. For example, Table 1. Anthropometric characteristics, MC, HRF, PA and WS performance

Response 1: Thank you for the suggestion. We have revised the formatting of all table titles and added a period after the table number (e.g., "Table 1.") to ensure consistency and to follow the standard academic style. We appreciate the reviewer’s attention to this detail, which helped improve the clarity and presentation of the manuscript. Lines 335, 359, 363, 367, 378.

Comments 2: Add the title of your background to the abstract.

Response 2: Thank you for your comment. The abstract has now been revised to include section headings, and “Background” has been explicitly added at the beginning to conform to the structured abstract format required by the journal. Lines 12-14.

Reviewer 2 Report

Comments and Suggestions for Authors

The authors have nicely addressed my concerns.

Author Response

Comments 1: The authors have nicely addressed my concerns.

Response 1: We would like to sincerely thank the reviewer for their constructive and insightful comments, which have significantly contributed to improving the quality and clarity of our manuscript. We carefully addressed each point raised and believe that the revised version of the article is stronger as a result.

Reviewer 3 Report

Comments and Suggestions for Authors

The modifications made by the authors have given the article the characteristics for a publication of high international prestige. Some of them, such as the inclusion of weak points in the work may in future help to improve procedures to achieve more reliable and valid results in the fiel of motor competence and key indicators in its development.

Other relevante aspects wich have been modified in this version focus on the variables associated with motor competence and its concrete manifestations. These events can serve as key elements for the development of educational programmes in the usual sports initiation processes and, above all, to provide indicators which specify the necessary elements for the study of health trends among young people. Whether o not tey are sportsmen.

As regards the procedure develop in the research (paragraph 2.6), mention should be made of the incorporation of measures taken by the researchers to minimise the influence of sequences on the order of tasks proposed in the results obtained by the participants.

Author Response

Comments 1: As regards the procedure develop in the research (paragraph 2.6), mention should be made of the incorporation of measures taken by the researchers to minimise the influence of sequences on the order of tasks proposed in the results obtained by the participants.

Response 1: 

We thank the reviewer for this helpful comment. We agree that the influence of task order should be addressed in more detail. We have therefore added the following sentence to clarify how we minimized the effect of task sequencing on participants' performance:

“To minimize potential order effects, children completed the tests in a randomized rotational order across small groups, and the more physically demanding tests (e.g., 20 m shuttle run and 4 × 10 m shuttle run) were consistently scheduled at the end or spaced apart by low-effort activities. This procedure helped to reduce fatigue and sequencing bias in the performance outcomes.” Lines 284-288.